# Fabrication and Characterization of Sn-Based Babbitt Alloy Nanocomposite Reinforced with Al_2_O_3_ Nanoparticles/Carbon Steel Bimetallic Material

**DOI:** 10.3390/ma13122759

**Published:** 2020-06-18

**Authors:** Mohamed Ramadan, Abdulaziz S. Alghamdi, Tayyab Subhani, K. S. Abdel Halim

**Affiliations:** 1College of Engineering, University of Ha’il, Ha’il P.O. Box 2440, Saudi Arabia; a.alghamdi@uoh.edu.sa (A.S.A.); k.abdulhalem@uoh.edu.sa (K.S.A.H.); 2Central Metallurgical Research and Development Institute (CMRDI), Cairo 11421, Egypt

**Keywords:** nanocomposite, nanoparticle, microstructure, mechanical, Babbitt, alumina

## Abstract

Sn-based Babbitt alloy was reinforced with alumina nanoparticles to prepare a novel class of nanocomposites. The route of liquid metallurgy in combination with stirring mechanism was chosen to prepare nanocomposites with three different loadings of alumina nanoparticles, i.e., 0.25 wt%, 0.50 wt% and 1.0 wt%. The molten mixture of metallic matrix and nanoparticles was poured over carbon steel substrate for solidification to manufacture a bimetallic material for bearing applications. The underlying aim was to understand the effect of nanoparticle addition on microstructural variation of Sn-based Babbitt alloy as well as bimetallic microstructural interface. The addition of 0.25 wt% and 0.50 wt% alumina nanoparticles significantly affected both the morphology and distribution of Cu_6_Sn_5_ hard phase in solid solution, which changed from needle and asterisk shape to spherical morphology. Nanocomposites containing up to 0.50 wt% nanoparticles showed more improvement in tensile strength than the one containing 1.0 wt% nanoparticles, due to nanoparticle-agglomeration and micro-cracks at the interface. The addition of 0.5 wt% nanoparticles significantly improved the wear resistance of Sn-based Babbitt alloy.

## 1. Introduction

In any engine, bearings are used to support moving mechanical parts, which protect them from frictional degradation. The importance of stronger bearing materials becomes many fold in powerful engines. Similarly, the significance of bearings in other mechanical systems such as pumps and turbines cannot be underestimated, and both the improved wear resistance and reliability of frictional components in terms of their mechanical strength become the prime requisites [1]. White metal is one of the bearing alloys that offers desired bearing surface properties in plain bearings; the common alloys have tin and lead as leading metals and are widely used in friction assemblies such as compressors, turbines, vehicles, and many other frictional units [2]. Babbitt alloys have two important lead-based and tin-based families. Generally poured bearing and insert bearings are made of Babbitt alloys; insert bearings have a backing of steel with inner layer made of Babbitt alloy. Lead-based Babbitt alloys have an outstanding low-load bearing characteristics. However, lead is a weak material and therefore tin, copper and antimony are added for improved strength [3].

One of the techniques to improve the mechanical and frictional properties of bearing materials is the incorporation of second phase as reinforcement. The resultant hybrid material named as composite provides tailored properties especially suitable for bearing applications [4]. After the arrival of nanotechnology, a novel class of composites emerged as nanocomposites containing reinforcements with one of their dimensions in nanometer scale [5,6]. Nano-reinforcements of zero, single and two-dimensions have been incorporated to prepare nanocomposites [7]. In addition to solitary reinforcement [8], dual reinforcements have also been incorporated at similar [9] and multiscale [10] levels. Among the three types of nanocomposites containing polymeric [11], ceramic [12] and metallic [13] matrices, comparatively little attention has been paid to nanocomposites containing ceramic and metallic matrices and still lesser to metallic matrices. In this context, the combination of traditional bearing materials with nano-reinforcements can lead to a novel class of nanocomposites ideally suitable for high performance bearing applications.

The two common processes appropriate for developing advanced metallic materials are: (a) new alloy formation and (b) novel heat-treatment process. The manufacturing of nanocomposites offers another route to prepare metal matrix composites containing nano-reinforcement of high strength and stiffness: typically ceramic materials. The nanoreinforcement of size down to nanometer-scale provides an opportunity to interact with dislocations and hence improve the mechanical properties of matrix material through novel strengthening mechanisms and thus remarkable improvement in mechanical properties [14].

A range of ceramic nano-reinforcements including alumina, silicon carbide, silicon dioxide, silicon nitride and zirconium dioxide have been incorporated into different metallic matrices such as aluminum, copper, magnesium, titanium and iron. Both liquid and solid-state processing routes have been utilized to prepare nanocomposites. Liquid processing routes offer better densification while solid-state routes provide better distribution of nano-reinforcement. To achieve optimum properties of nanocomposites, both near-theoretical density and uniform dispersion of nano-reinforcements are fundamental requirements. Liquid metallurgy in combination with stirring technique offer better dispersion while solid-state processing primarily powder metallurgy in combination with pressure-assisted techniques such as hot-pressing and spark plasma sintering provides better densification. The low wettability of ceramic nano-reinforcements with metallic matrices leads to their poor dispersion, which can be improved by chemical and mechanical routes. In addition to the ex-situ incorporation of nano-reinforcements, in-situ formation of nano-reinforcements has also been investigated. Finally, friction stir processing is another solid-state processing route to prepare nanocomposites with improved mechanical performance.

Among a variety of metallic matrices, aluminum and its alloys are widely used for preparing nanocomposites, and scarce attention has been paid to improve the mechanical properties of bearing materials by adopting the technique of composite preparation. Recently, carbon nanotubes have been added in white metal (Babbitt), which increased its wear resistance to almost an order of magnitude [15].

In the present work, alumina nanoparticles have been incorporated in Sn-based Babbitt alloy through stir casting technique. The developed nanocomposite was bonded with carbon steel substrate to prepare a bimetallic material for bearing applications. The underlying aim was to investigate the effect of ceramic nanoparticles addition upon the microstructural evolution and resultant mechanical property improvement especially tribology. Another objective was to observe the microstructural interface of nanocomposites with carbon steel substrate.

## 2. Materials and Methods

Alumina nanoparticles of spherical morphology and average particle size of ~50nm were used as nano-reinforcement, as procured from Metkon Technology. Tin-based Babbitt alloy (Rotometals, San Leandro, CA, USA) was used as the matrix material to prepare nanocomposites. To prepare the bimetallic material, carbon steel was used for bonding with nanocomposites. The chemical compositions of Sn-based Babbitt alloy and carbon steel (local manufacturing Co., Riyadh, Saudi Arabia) are given in Table 1. Chemical composition of Sn- Babbitt alloy shown in Table 1 as per conducting by manufacturing company (Rotometals). For carbon steel substrate, ASCert metal scan analyzers (Arun Technology, Crawley, UK) was used for chemical analysis.

A charge of 400 g of tin-based Babbitt alloy was placed in a graphite crucible and heated in electrical furnace to a temperature of 375 °C (heating rate = 9.4 °C/min.). Subsequently alumina nanoparticles were added after lowering the temperature to 330°C (cooling rate = 5.6 °C/min.) in a semi-solid state of alloy. Alumina nanoparticles were wrapped in aluminum foil and were directly immersed in small packages. The stirring operation was performed continuously during the addition of nanoparticles. The semi-solid slurry was then heated to 420 °C so that a fully liquid state could be achieved before pouring. The stirring in semi-solid state was performed at a speed of 800 rpm for 10 min, which was followed by stirring at 200 rpm until pouring.

The stirring was carried out mechanically using a two-blade impeller. After mixing and stirring stage, the molten metal was poured on a tinned carbon steel substrate, which was inserted into a pre-heated metallic mold at 150 °C. The carbon steel substrate had a thickness of 4mm and diameter 59 mm. The schematic of the process is presented in Figure 1 and Figure 2 along with the images of the furnace, stirring mechanism, metallic mold and carbon steel substrate. Tinning process of carbon steel substrate involves the deposition of metallic tin and flux mixture on the surface of steel, as explained elsewhere [16].

Nanocomposites of three types were prepared after incorporating three loading fractions of alumina nanoparticles in tin-based Babbitt alloy, i.e., 0.25 wt%, 0.50 wt% and 1.0 wt%. A reference sample without the addition of alumina nanoparticles was also prepared. For preparing bimetallic sample, the same type of carbon steel was used.

Optical microscopy was performed to observe the microstructure of bimetallic specimens containing nanocomposite and carbon steel alongwith their interface. The specimens were cut, ground, polished and etched with a solution of 4% nital (96% ethical alcohol and 4% nitric acid HNO_3_). Optical microscope fitted with digital camera (Olympus GX51, Tokyo, Japan) was used for microstructural investigations at magnifications of 200× and 500×). For electron micrsocopy, field emission gun scanning electron microscope (FEG-SEM, FEI, Eindhoven, The Netherlands) was utilized by a Quanta 250 FEG, (Eindhoven, The Netherlands) with energy-dispersive X-ray spectroscopy (EDS) was used. Interface between the nanocomposites and substrate steel was observed in electron microscopy and compositional analysis was performed using EDS. Moreover, SEM and EDS was also performed on the worn surfaces after wear tests, as discussed further below.

The hardness of nanocomposites was tested using Vickers microhardness testing machine (VLS 3853, Shimadzu, Japan). The hardness testing was performed at the load of 1kgf for the dwell time of 5 sec. The specimens were polished until the surface finish of 1 µm. At least five readings of each of the nanocomposite were taken for the average value of hardness.

The tensile and shear testing until complete failure of the bond were conducted using tensile testing machine Instron 5969 characterized by a maximum load of 50 KN. The interfacial shear strength was measured by a tensile shear method. The dimensions of tensile shear specimen are reported in a previous work [17].

To evaluate the wear resistance of nanocomposites, pin-on-ring wear test was performed. The fixed samples of nanocomposites were tested against the rotating steel wheel at a constant speed of 260 rpm for 10 min under a constant loading of 28N.The wear resistance of the samples was measured in terms of the weight-loss.

## 3. Results

### 3.1. Microstructure of Reinforced Sn-based Babbitt Alloy and Bimetal Interface

Microstructures of both of the materials used in the fabrication of tin-based Babbitt alloy/carbon steel bimetallic composite are shown in Figure 3 while Figure 4 shows SEM microstructures (Figure 4a,c) and EDS analysis (Figure 4b,d) of Cu_6_Sn_5_ hard phase and matrix of tin-based Babbitt alloy of bimetallic material. Figure 3a shows an optical microstructure of Sn-based Babbitt alloy revealing Cu_6_Sn_5_ and SnSb phases reinforced in solid solution of SnSbCu matrix. Cu_6_Sn_5_ phase has the shape of needles and asterisks. The microstructure of a matrix of solid solution with hard embedded phases of Cu_6_Sn_5_ and SnSb make a metal matrix composite possessing fatigue resistant properties [17,18,19]. Previous studies [20,21,22] reveal that the variation in the chemical composition of tin-based Babbitt alloy modifies the microstructure. Alloys containing less than 8% Sb are characterized by a solid solution with distributed needles of Cu_6_Sn_5_, copper-rich constituents and fine precipitates of SbSn. On the contrary, the alloys containing greater than 8% Sb exhibit a primary cuboid phase of SbSn. The microstructure of Sn-based Babbitt alloy acquired in the present study is in good agreement with those available in literature [20,21,22]. The microstructure of low carbon steel substrate (Figure 3b) shows the two very common phases of ferrite and pearlite; the mutual percentage of the presence of two phases verifies the carbon content in the steel.

Figure 5 shows the microstructures of Sn-based Babbitt alloy without (Figure 5a) and with three different loadings of alumina nanoparticles, i.e., 0.25 wt%, 0.50 wt%, and 1.0 wt%, correspondingly (Figure 5b–d). The addition of alumina nanoparticles significantly affected both the morphology and distribution of Cu_6_Sn_5_ hard phase in solid solution. It is evident that Cu_6_Sn_5_ phase changed from needle and asterisk shape to spherical morphology after the addition of alumina nanoparticles. Pervious study [23] reported that addition of Al_2_O_3_ nanocomposites to hyper-eutectic Al-Si alloy, change the morphology of primary Si to a typical polyhedral shape. A possible reason of the change in the morphology of Cu_6_Sn_5_ phase from needle to spherical shape may be the hindrance in the growth due to the presence of alumina nanoparticles. However, a trend of Cu_6_Sn_5_ phase agglomeration was observed as the alumina nanoparticle content increased to 1.0 wt% loading where clearly identifiable regions of Cu_6_Sn_5_-rich and Cu_6_Sn_5_-depleted phases can be seen (Figure 5d). It may be due to the reason that at 1.0 wt% content, alumina nanoparticles could not be dispersed uniformly and at alumina-rich regions, the growth of Cu_6_Sn_5_ phase was very low and vice versa.

The microstructures of interfaces of bimetallic samples containing tin-based Babbitt alloy with and without the addition of alumina nanoparticles and substrate steel are shown in Figure 6. A visual observation of the interfacial microstructures of bimetallic interface shows a good bonding between the two metallic parts, i.e., nanocomposites of tin-based Babbitt alloy matrix containing alumina nanoparticles and carbon steel substrate. A uniform interfacial bonding is observed, which shows that the molten nanocomposite solidified on the surface of steel substrate in such way that no interfacial defects were noted. Tin-based Babbitt alloy with 0.25 wt% and 0.50 wt% additions of alumina nanoparticles revealed a good quality of interfacial bonding compared with 1 wt% addition.

Interfacial micro-voids of tin-based Babbitt alloy with and without alumina nanoparticles were observed, as shown in Figure 7. EDS analysis of the micro-voids show the presence of all elements as found in both metals (Figure 7b). It was reported [24] that hot tearing resistance of the A206/1 wt% Al_2_O_3_ nanocomposite was significantly better than that of the pure A206 alloy. The nanocomposite containing 1.0 wt% alumina nanoparticles showed micro-crack (Figure 7d), which is another indication of the presence of alumina nanoparticle agglomerates. In case the nanoparticles were not completely wetted by the surrounding matrix material, these acted as defects and raised the localized stress level thus initiating cracks, as observed in Figure 7d. However, no such indication was noted without the addition of nanoparticles (Figure 7a) and with the incorporation of 0.50 wt% nanoparticles (Figure 7c).

Figure 8 shows the SEM image and EDS analysis of nanocomposite containing 0.50 wt% alumina nanoparticles. The EDS area analysis shows the presence of Al and O atoms in both the Cu_6_Sn_5_ phase and Sn-based Babbitt matrix. It is clear that the concentration of Al and O in Babbitt matrix is relatively higher than that found in Cu_6_Sn_5_ phase. The EDS area analysis shows that the presence of Al_2_O_3_ nanoparticles in Babbitt matrix is relatively high compared to that found in Cu_6_Sn_5_ phase. The modification in Cu_6_Sn_5_ phase may be attributed to the result of the effect of pushed γ-Al_2_O_3_ on restricting the growth of the particles during solidification into remaining liquid.

### 3.2. Mechanical Properties of Reinforced Sn-based Babbitt Alloy and Bimetal Interface

The hardness values of three nanocomposites and the reference sample of Sn-based Babbitt alloy are shown in Figure 9 and Table 2. It can be seen that the hardness of Sn-based Babbitt material was improved after the addition of alumina nanoparticles up to 0.5 wt%. In contrast, a minor decrease in hardness was noticed with rising content of nanoparticles up to 1.0 wt%. The modification in the morphology of hard Cu_6_Sn_5_ phase and its uniform distribution is the reason of high hardness up to 0.5 wt%. Otherwise, it can be inferred that the rising effect in hardness in nanocomposites was nullified by the gradual agglomeration of nanoparticles in their increasing content. A secondary effect may be due to the change in the morphology of hard Cu_6_Sn_5_ phase along with the presence of Cu_6_Sn_5_ phase agglomerates [23]. Nevertheless no significant decline in hardness profile was observed, which could otherwise be improved by increasing the quality of nanoparticle dispersion keeping in view the large difference in the hardness values of metallic tin-based Babbitt alloy and alumina.

The average tensile strength values of nanocomposites containing three different nanoparticle loadings and corresponding shear strength values of bimetallic interfaces are shown in Figure 10 and Table 2. The addition of nanoparticles increased the strength up to 68 MPa with 0.50 wt% loading showing a rise of 21% in comparison to reference as-cast matrix. However, a further rise in nanoparticle addition up to 1.0 wt% reversed the rising trend and a decrease in strength was witnessed. A similar trend was observed in interfacial shear strength of the three nanocomposites after preparing bimetallic materials with carbon steel, i.e., maximum strength was observed at 0.50 wt% loading which decreased by further addition.

### 3.3. Wear Properties of Reinforced Sn-based Babbitt Alloy

Figure 11 shows the variation in weight-loss against the loading fraction of alumina nanoparticles in Sn-based Babbitt alloy. A value of 2.1 ± 0.1 mg was noted in unloaded sample of tin-based Babbitt alloy, which dramatically decreased to 1.30 ± 0.07 mg after adding 0.25 wt% alumina nanoparticles, which decreased further to 1 ± 0.05 mg at 0.5 wt% alumina nanoparticles. However, a regain in the value was noted by increasing the nanoparticle content to 1.0 wt%, i.e., 1.82 ± 0.09 mg. It can be seen that the addition of nanoparticles up to the values of 0.50 wt% decreased the weight-loss but further rise in the content of nanoparticles led to their agglomeration, which increased the weight-loss. Indeed poor dispersion of nanoparticles produces defects, which are actually the unfilled matrix areas between nanoparticles. These unfilled matrix areas or the space between particles is in fact the inability of the matrix to encapsulate the nano-reinforcement. Each nanocomposite manufacturing technique has a limitation in the loading fraction of the nano-reinforcement. In the present work, the stirring parameters together with the associated manufacturing setup provided a maximum nanoparticle dispersion of 0.5 wt%. As a result, the wear properties were also found optimum at the same loading fraction of nanoparticles.

Figure 12 shows the SEM images of the worn surfaces of four types of specimens with and without nanoparticle addition along with the EDS results of unloaded sample at two points. The results of EDS analysis shown in Figure 4e,f confirmed that the proper peaks of elements coming from the SnSbCu matrix are clearer for two selected areas of the worn surfaces of Sn-based Babbitt alloy without Al_2_O_3_ nanoparticles addition. A lower content of Sb and Cu, as well as other additives nanoparticles in matrix, directly influences the hardness, as well as tribological properties of this area of Sn- based Babbitt structure. No cracks were observed on the wear tracks formed on the surfaces of the specimens. The grooves can be seen parallel to the sliding direction. The maximum wear was observed in the unloaded specimen, which may be due to large plastic deformation owing to low hardness value. With the systematic increase in hard nanoparticles in the matrix, the mechanism of wear shifted from adhesive to abrasive wear. As the contacting surface is steel during the wear test, therefore, an adhesion of tin-based Babbitt alloy was seen in the specimen without alumina nanoparticles, which shifted to abrasion in the presence of nanoparticles in specimens up to 1.0 wt%. The worn surfaces also became gradually smoother in the presence of nanoparticles. However, in specimen containing 1.0wt% nanoparticles, the presence of defects resulted in poor density and, hence, the increased rate of material removal though the evidence of abrasive wear is evident.

The decrease in weight-loss in the wear test may be attributed to the change in the morphology of Cu_6_Sn_5_ phase from asterisk and needle shape to a spherical shape. A spherical shape is always smoother than the asterisk and needle shape and may have the ability to wear less in comparison to the other two shapes. During the wear test, the contacting surface may slip over the spherical shape in comparison to breaking the edges of needle and asterisk shapes. As evidence, tin-based Babbitt alloy exhibited the maximum weight-loss, which decreased with the addition of alumina nanoparticles, which changes the shape of Cu_6_Sn_5_ phase from asterisk and needle shape to spherical shape. However, the effect of the change in the morphology of Cu_6_Sn_5_ phase was nullified due to their agglomeration at 1 wt% alumina nanoparticle loading. The Cu_6_Sn_5_-rich and Cu_6_Sn_5_-depleted regions clearly developed identifiable soft and hard regions. Soft regions may have shown more wear and hard regions may have exhibited more pull-out phenomenon. At the same time, the loading of 1 wt% alumina nanoparticles may also have developed nanoparticle-agglomerates, which also adversely affected the wear process. Nevertheless, the minimum wear and weight-loss was observed at the loading of 0.50 wt% alumina nanoparticles, which is highly desirable for bearing applications.

## 4. Discussion

The acquired results show that alumina nanoparticle loadings of 0.25 wt% and 0.50 wt% increase the hardness, tensile strength, and wear resistance of Sn-based Babbitt alloy, which decrease by further addition in nanoparticles. A possible reason may be the misfit effect in the lattice parameters of the matrix phase, which may be due to the result of cooling induced changes caused by the difference between the coefficients of thermal expansion between the Cu_6_Sn_5_ phase and the nanoparticles, thus resulting in an increase in the hardness [25]. Moreover, enhanced dislocation generation and reduced Cu_6_Sn_5_ phase owing to the presence of the nanoparticles may be contributing to the increased hardness of the Sn-based Babbitt matrix. In addition to mismatch effect in thermal expansion, the increase in the hardness is also mismatch in elastic moduli of the nanoparticles. 

It has been shown [26] that the addition of the alumina nanoparticles results in a constraint in the plastic flow of matrix. This matrix can flow only with the movement of nanoparticles or over the particles during plastic deformation. The matrix shows a constrained plastic deformation because of smaller inter-particle distance and results in improvement in the flow stress. Finally, the strengthening mechanisms that are well-documented for coarse-grained metals and alloys can also operate in nano-crystalline materials after being modified by nano-scale particles and non-equilibrium microstructures [27]. It is suggested that the alumina nano-particles contributed to Orowan hardening. Hence, nano-dispersed cast structures carry both the nature of composites and refined Sn-based Babbitt structures. It is therefore important at this stage to understand the active strengthening mechanisms in nanoparticle-reinforced Sn-based Babbitt materials, as this could be the key to developing novel microstructures with improved properties.

## 5. Conclusions

A novel class of nanocomposites containing alumina nanoparticles in Sn-based Babbitt alloy was prepared with nanoparticle additions of 0.25 wt%, 0.50 wt%, and 1.0 wt%. It was found that the addition of nanoparticles changed the morphology of Cu_6_Sn_5_ phase from needle and asterisk shape to a spherical form. A parallel trend of Cu_6_Sn_5_ phase agglomeration was also observed while increasing nanoparticle content leading to identifiable Cu_6_Sn_5_-rich and Cu_6_Sn_5_-depleted areas. Sn-based Babbitt alloy with 0.25 wt% and 0.50 wt% nanoparticles revealed improvements in the tensile strength and quality of interfacial bonding, while the one containing 1.0 wt% nanoparticles showed lower tensile and shear strength values due to nanoparticles agglomerations in the matrix and micro-cracks at the interface. A substantial decrease in weight-loss was observed in the wear test by adding 0.5 wt% nanoparticles, which otherwise increased at 1.0 wt% loading. The developed bimetallic material containing 0.50 wt% alumina nanoparticles in Sn-based Babbitt alloy bonded with carbon steel substrate may be considered as a promising material for high-performance bearing applications.

## Figures and Tables

**Figure 1 materials-13-02759-f001:**
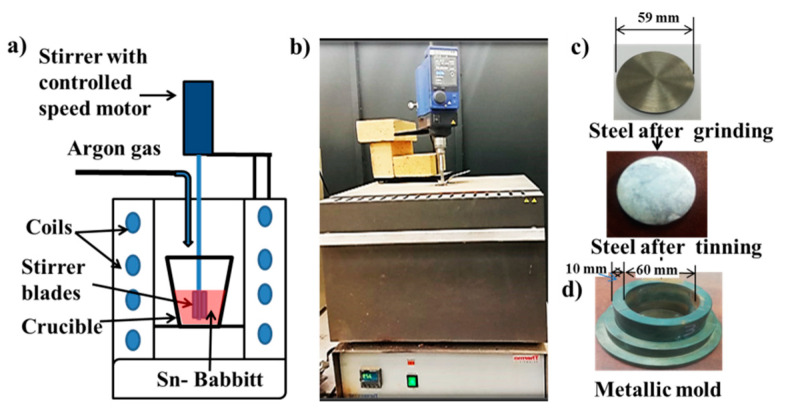
Furnace and mold used for preparing nanocomposite containing tin-based Babbitt alloy reinforced with alumina nanoparticles and carbon steel specimen before and after tinning, (**a**) Schematic drawing for furnace and stirrer, (**b**) photocopy of furnace and stirrer, (**c**) carbon steel substrate and (**d**) metallic mold.

**Figure 2 materials-13-02759-f002:**
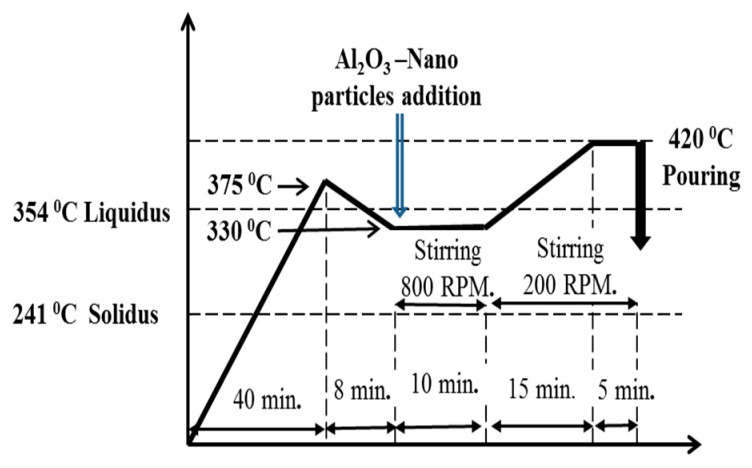
Sequence of preparing nanocomposite of tin-based Babbitt alloy reinforced with alumina nanoparticles by stir-casting process.

**Figure 3 materials-13-02759-f003:**
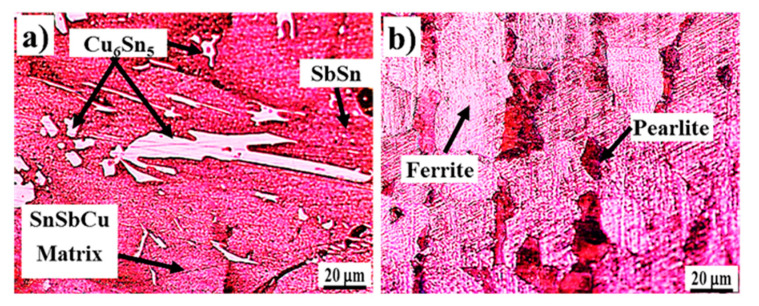
Microstructures of (**a**) tin-based Babbitt alloy, and (**b**) steel substrate, used for preparing bimetallic casting.

**Figure 4 materials-13-02759-f004:**
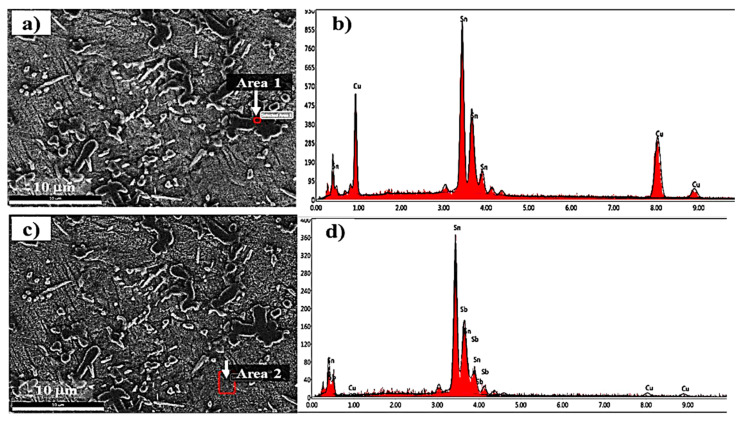
(**a,c**) SEM images of tin-based Babbitt alloy and (**b,d**) graphical presentation of EDS microanalysis of precipitates (Area 1) and matrix (Area 2) in Sn-based Babbitt alloy.

**Figure 5 materials-13-02759-f005:**
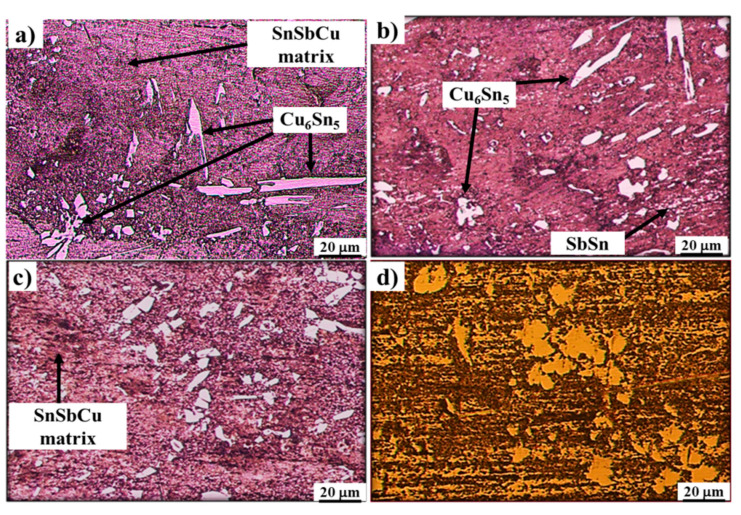
Microstructures of (**a**) tin-based Babbitt alloy, and tin-based Babbitt nanocomposites containing (**b**) 0.25 wt%, (**c**) 0.50 wt%, and (**d**) 1.0 wt% alumina nanoparticles correspondingly.

**Figure 6 materials-13-02759-f006:**
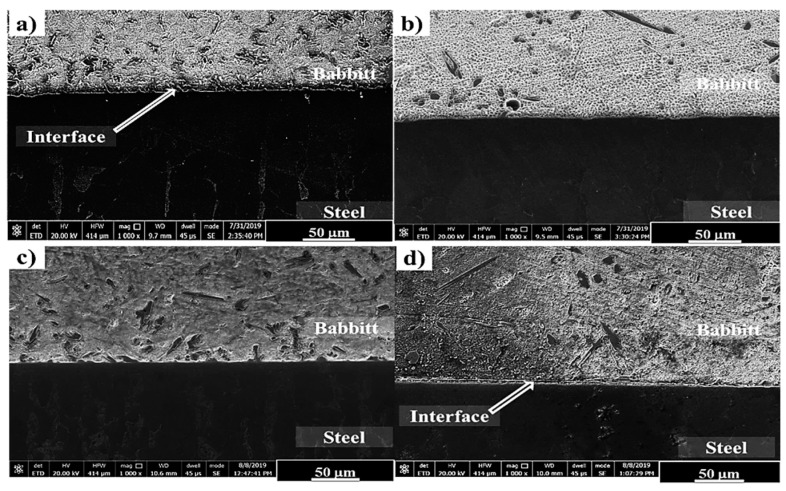
SEM images of interfaces in tin-based Babbitt/steel bimetallic cast specimens with different loading fraction of alumina nanoparticles in tin-based Babbitt alloy: (**a**) 0 wt %, (**b**) 0.25 wt%, (**c**) 0.50 wt% and (**d**) 1.0 wt%.

**Figure 7 materials-13-02759-f007:**
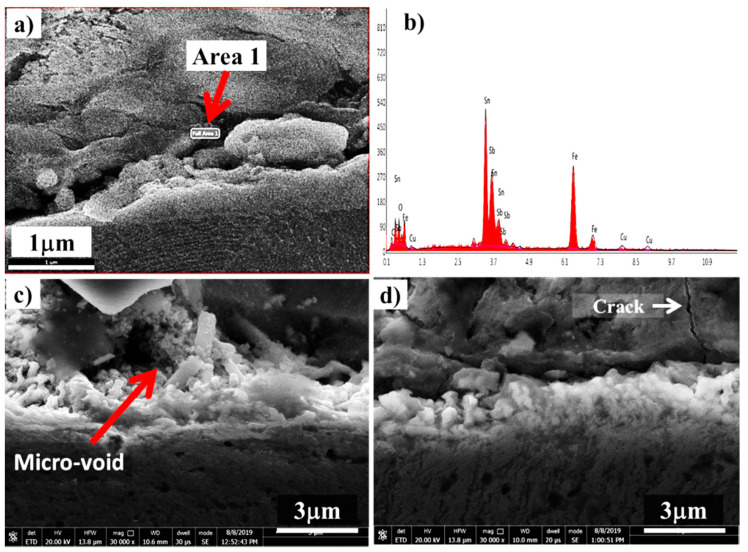
SEM images of interfaces in tin-based Babbitt/steel bimetallic specimens with different loading fractions of alumina nanoparticles in tin-based Babbitt alloy: (**a**) 0 wt%, (**c**) 0.50 wt% and (**d**) 1.0 wt%. (**b**) EDS of tin-based Babbitt interface with 0 wt%.

**Figure 8 materials-13-02759-f008:**
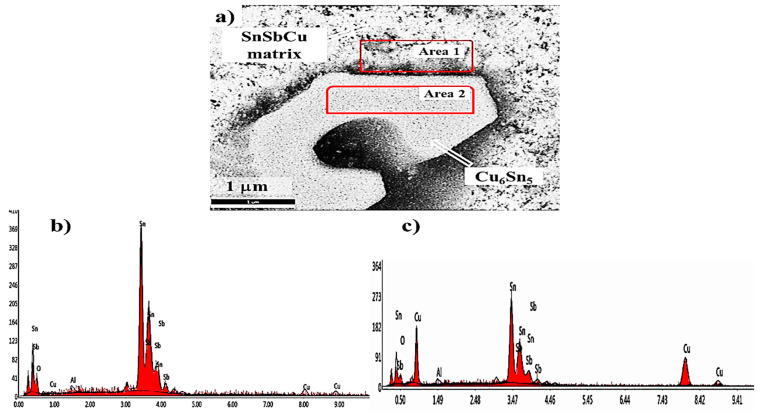
(**a**) SEM image of tin-based Babbitt alloy nanocomposite containing 0.5 wt% alumina nanoparticles, (**b**) and (**c**) graphical presentation of EDS results at two selected areas of matrix and Cu_6_Sn_5_ Phase consequently.

**Figure 9 materials-13-02759-f009:**
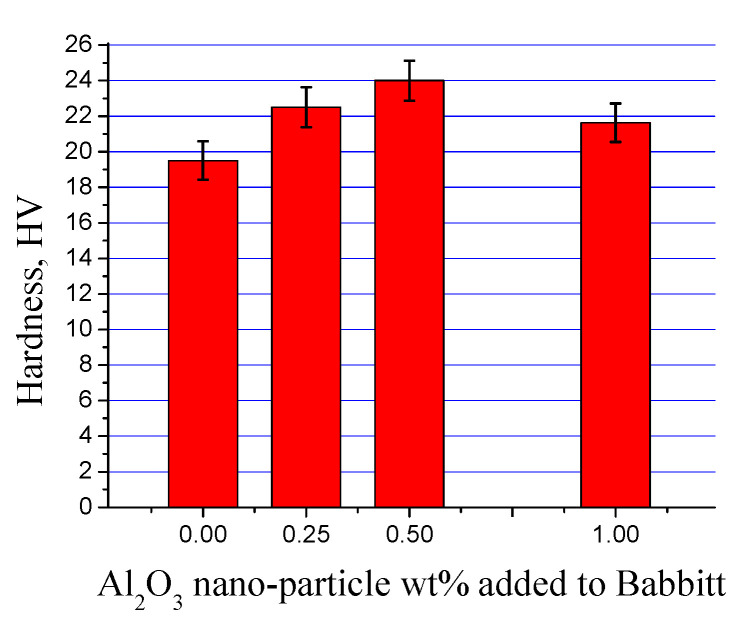
Hardness values of tin-based Babbitt alloy nanocomposite containing alumina nanoparticles.

**Figure 10 materials-13-02759-f010:**
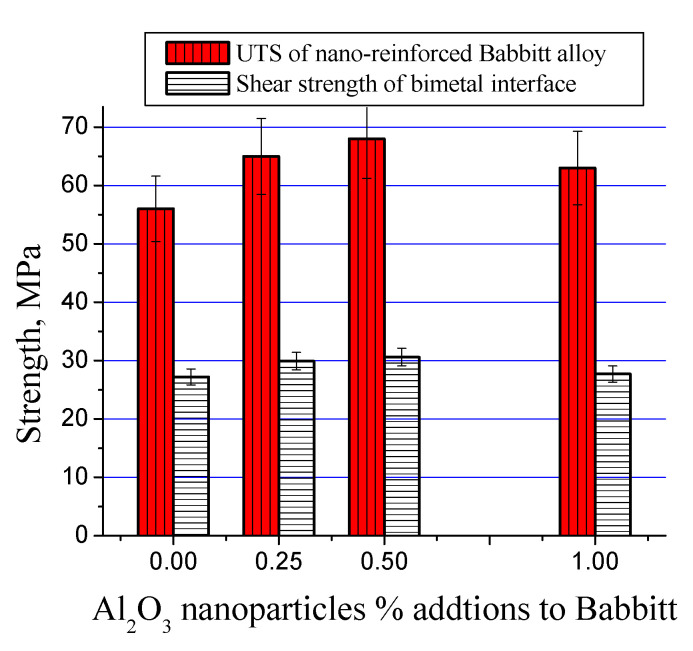
The average tensile strength values of nanocomposites containing alumina nanoparticles and average shear strength values of corresponding bimetallic interfaces.

**Figure 11 materials-13-02759-f011:**
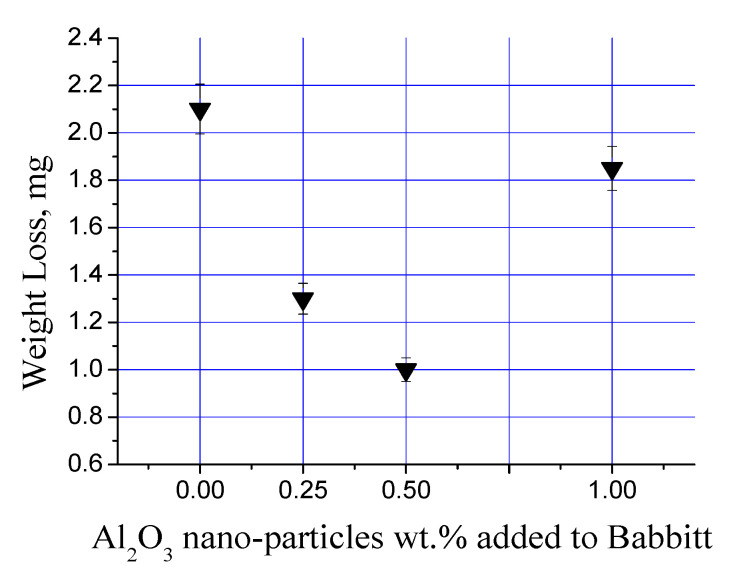
The average wear results of tin-based Babbitt alloy reinforced with alumina nanoparticles.

**Figure 12 materials-13-02759-f012:**
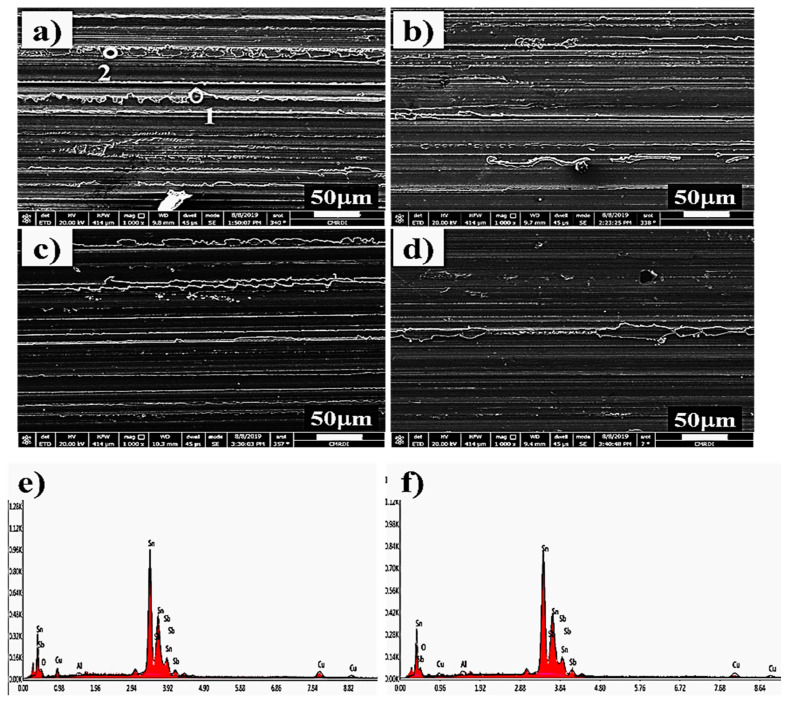
SEM images of worn surfaces of tin-based Babbitt alloy nanocomposites containing (**a**) 0 wt%, (**b**) 0.25 wt%, (**c**) 0.5 wt% and (**d**) 1.0 wt% alumina nanoparticles, and (**e,f**) EDS results of two selected areas of worn surfaces Sn-based Babbitt alloy without Al_2_O_3_ nanoparticles addition.

**Table 1 materials-13-02759-t001:** Chemical compositions of tin-based Babbitt alloy and carbon steel substrate.

Materials	Chemical Compositions (wt.%)								
	Sb	Cu	Pb	C	Si	Mn	Cr	Sn	Fe
Tin Babbitt Alloy	6.5–7.5	2.5–3.5	0.35	-	-	-	-	Bal.	
Carbon Steel	-	0.20	-	0.14	0.30	0.48	0.14	-	Bal.

**Table 2 materials-13-02759-t002:** Tensile strength, hardness and shear strength of tin-based Babbitt/carbon steel bimetal with without Al_2_O_3_ nanoparticles additions.

S/N	Composition	Sn-Babbitt Zone		Interface Zone
		Tensile Strength, MPa	Hardness, HV	Shear Strength, MPa
1	Sn-Babbitt/steel bimetal	56 ± 5.6	19.5 ± 1.08	27.2 ± 1.36
2	Sn-Babbitt + 0.25% Al_2_O_3_ nanoparticles/steel bimetal	65 ± 6.5	22.5 ± 1.13	29.9 ± 1.50
3	Sn-Babbitt + 0.50% Al_2_O_3_ nanoparticles/steel bimetal	68 ± 6.8	24.0 ± 1.13	30.6 ± 1.53
4	Sn-Babbitt + 1.00% Al_2_O_3_ nanoparticles/steel bimetal	63 ± 6.3	21.6 ± 1.08	27.7 ± 1.39

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
