# Peer review of "Fabrication and Characterization of Sn-Based Babbitt Alloy Nanocomposite Reinforced with Al2O3 Nanoparticles/Carbon Steel Bimetallic Material"

_materials, 2020, doi:10.3390/ma13122759_

Round 1

Reviewer 1 Report

For SEM images of interfaces in tin-based Babbitt/steel bimetallic cast specimens with different loading fraction of alumina nanoparticles (Figure 6), I wish recommended the realization of the SEM images at the same scale.
In the graphical representation of mechanical properties, I suggest the deleting on the x axe (Al2O3 nano-particle wt%) of the percent 0.75 % Al2O3 because this sample was not be analyzed.

Author Response

For SEM images of interfaces in tin-based Babbitt/steel bimetallic cast specimens with different loading fraction of alumina nanoparticles (Figure 6), I wish recommended the realization of the SEM images at the same scale.
In the graphical representation of mechanical properties, I suggest the deleting on the x axe (Al2O3 nano-particle wt%) of the percent 0.75 % Al2O3 because this sample was not be analyzed.

SEM images in Figure 6 are now at the same scale, as recommended. 0.75% Al2O3 loading on the x-axis has been removed, as suggested.

Reviewer 2 Report

This study aims at investigating the behavior of Sn-based Babbitt alloy with the addition of alumina nanoparticles and also the fabrication of a bimetallic material (Babbitt alloy on a carbon steel substrate). The microstructure is investigated, as well as tensile strength, hardness and wear resistance. In the abstract a short presentation on the results and improvements of the reinforced Babbitt alloy is presented. In the introduction, the concept of nano-composites is introduced, followed by the processes used to develop advanced metallic materials and then a quick review over the ways of making ceramic nano-reinforcements. Then, the method used in order to fabricate the alloy Babbitt nanocomposite is described in detail. After that follows the presentation of the results, which are also shown in enough detail and are scientifically well explained, thought a few more references should be added (see bellow).

Some remarks follow:

  • Some references is advised to be added in (and around) lines 168, 192, 213 and the first paragraph in the discussion section.
  • In chapter 3.2, a table of properties etc values is highly recommended to be added.
  • In figure 11, the line connecting the points shouldn’t be there, since it is an assumption that right after point 0.5 % wt the weight lose increases, since it may rather decreases further before it goes up.
  • In figure 11, either the error bars are false or the error values in lines 237 to 240 are false.
  • In lines 180 – 181 it is reported that the interfacial bonding quality is the same in the samples with and without nanoparticles. On the contrary, in lines 310 – 311 it is stated that the samples with 0.25 % wt and 0.5 % wt show better interfacial bonding quality.
  • In the whole manuscript, it must be clarified that the change on the value of a property etc from 0.5 % wt to 1.0 % (either increase or decrease) is between 0.5 % and 1.0 % and not the pristine 0 %.
  • In the whole manuscript:
    • Add blank between number, wt and %.
    • Add blank before and after +- symbol.
  • Please change the following
    • Line 30, change becomes to become.
    • Line 42, change range to scale.
    • Line 116, change mi(c)roscopy to microscopy.
    • Line 123, “Moreover.” replace full stop (.) to coma (,).
    • Line 179, remove “a”.
    • Line 203, change “phase( )may and Cu6Sn5( )phase” to “phase may and Cu6Sn5 phase”
    • In figure 8, add description for the image “c” in the citation.
    • In figure 6, in b and d the “Babbitt” isn’t clearly visible. Also, please correct the percentage to 0.25 and 0.5 from 2.5 and 5.0. Same for image 7.
    • In figure 4, a and c the scale bars aren’t visible.

Author Response

·         Some references is advised to be added in (and around) lines 168, 192, 213 and the first paragraph in the discussion section.

References have been added in and around lines 168, 192, 213 and the first paragraph of discussion section.

·         In chapter 3.2, a table of properties etc values is highly recommended to be added.

·         A table of properties in section 3.2 has been added, as recommended.

·          

·         In figure 11, the line connecting the points shouldn’t be there, since it is an assumption that right after point 0.5 % wt the weight lose increases, since it may rather decreases further before it goes up.

·         In Figure 11, the connecting line has been removed, as suggested.

·          

·         In figure 11, either the error bars are false or the error values in lines 237 to 240 are false.

·         The error values and corresponding error bars in Figure 11 have been corrected, as correctly pointed out.

·          

·         In lines 180 – 181 it is reported that the interfacial bonding quality is the same in the samples with and without nanoparticles. On the contrary, in lines 310 – 311 it is stated that the samples with 0.25 % wt and 0.5 % wt show better interfacial bonding quality.

·         The sentences about the interfacial quality have been corrected and properly placed.

·          

·         In the whole manuscript, it must be clarified that the change on the value of a property etc from 0.5 % wt to 1.0 % (either increase or decrease) is between 0.5 % and 1.0 % and not the pristine 0 %.

·         As suggested, the change in the value of property has been correctly related to samples.

·          

·         In the whole manuscript:

·         Add blank between number, wt and %.

·         Added

·         Add blank before and after +- symbol.

·         Added

·         Please change the following

·         Line 30, change becomes to become.

·         Changed from becomes to become

·          

·         Line 42, change range to scale.

·         Changed from range to scale

·          

·         Line 116, change mi(c)roscopy to microscopy.

·         Changed from miroscopy to microscopy

·          

·         Line 123, “Moreover.” replace full stop (.) to coma (,).

·         Comma has been added after moreover instead of full stop.

·          

·         Line 179, remove “a”.

·         “a” has been removed.

·          

·         Line 203, change “phase( )may and Cu6Sn5( )phase” to “phase may and Cu6Sn5 phase”

·         Corrected as suggested.

·          

·         In figure 8, add description for the image “c” in the citation.

·         Description for the image “c” has been done.

·          

·         In figure 6, in b and d the “Babbitt” isn’t clearly visible. Also, please correct the percentage to 0.25 and 0.5 from 2.5 and 5.0. Same for image 7.

·         In Figure 6, in b and d, the Babbitt is now clearly visible. Samples have been correctly written in Figures 7 and 8.

·          

·         In figure 4, a and c the scale bars aren’t visible.

·         Scale bars in Figure 4 are now clearly visible.

Reviewer 3 Report

1. There are many references on the same author (Khan), I offer to expand the search. 2. What is the speed with which heating  conducted under stirring and have you varied the time? 3. Can you perform EDX analysis not point-to-point but in areas to estimate the distribution of alumina? 4. You write the 0.50 wt% then 0.5 %, bring to the general view. 5. Have there been studies on attrition and tear over time (lifetime)?

Author Response

1. There are many references on the same author (Khan), I offer to expand the search. 2. What is the speed with which heating conducted under stirring and have you varied the time? 3. Can you perform EDX analysis not point-to-point but in areas to estimate the distribution of alumina? 4. You write the 0.50 wt% then 0.5 %, bring to the general view. 5. Have there been studies on attrition and tear over time (lifetime)?

1. The search has been expanded and new references have been added. 2.  Added in the text of the manuscript. 3. At this stage, it is difficult. We will do it in our next publication. 4. We will bring it to a general view. 5. It was out of the scope of the present research work.

Reviewer 4 Report

The presented to Materials journal manuscript 824064 is an interesting work about fabrication of Babbitt alloy nanocomposites as potential candidates for bearings applications.

The introduction is relevant, the literature analyses is used through the manuscript providing explanations of the obtained results and a discussion. In spite of low quality of some images, that should be certainly improved, manuscript is presented in a good and easy to read way.

I would recommend this manuscript to be published in Materials journal after some minor corrections. Here they are:

Lines 84-85. Could you add a manufacturer (Company and a country) of the Babbitt alloy and a carbon steel? Where did you get quantitative composition of alloys? Has it come from a manufacturer already?

Lines 92-93. What was the speed of heating (from room temperature to 375°C, etc.) /cooling (from 375°C to 330°C) stages?

Lines 99-100. Please, indicate parts of your Figure 1, as a, b and c. Add scales to the images or indicate sizes, where it is possible.

Line 118. What are the concentrations of acids? Were they concentrated? Please, specify it.

Line 121. What is the manufacturer and the model of the electron microscope? Please add this information.

Line 122. Should be. “Interface … was observed…”

Line 123. Should be a comma instead of a dot after the word “Moreover”

Line 155. Horizontal composition of images would be more appropriate.

Lines 157-158. The scales on images and numbers, as well as elements on graphs are not readable. Please change the quality of this Figure. Otherwise, it is not clear what is difference between images a) and c), as both of them related to tin-based Babbitt alloy.

Line 164. Please, add word “correspondingly” in the end and verify the letters – a, b, c, and d in th text and in the figure caption (line 161). it seems as an error, i.e. should be “Figure 5b, c and d, correspondingly”.

Line 172. Should be “… did not dispersed…”

Line 184. What are the loading fractions: 0.25, 0.5 and 1.0 wt.% or 2.5, 5.0 and 1.0 wt.%? Is it a mistake?

Line 188. What is in the image c)? Please, correct the caption. The same question about loading fraction as for Figure 6, is it 5.0 wt.% or 0.5 wt.%?

Figure 6 and 7. The scale bars on SEM images are very small. Please, increase them or modify the images.

Line 200. Please, explain better from where goes that Al and O content is bigger in the Babbitt matrix than in the Cu6Sn5 phase? Is it possible to add a table with elements content?

Line 202 and 203. Some spaces are missed between the words.

Line 207. Figure 8 has very bad scale bar. Please, improve it as well as EDS graphs, where the chemical elements are seen with difficulties. Add explanation for c) in the figure caption. Is b) is for the area 1 and c) for the area 2?

Line 254. Figure 12. The same problem of scale bars and chemical elements labels. It should be definitely changed to better ones. Please, indicate the correspondence between e) and f) and areas of EDS analyses.

Lines 257 and further. What was the need for EDS analysis of tin-based Babbitt alloy here? These results are not discussed in the text. Please, describe them or remove.

Plese, add spaces between numbers and words, also after some braquetes of references and punctuation signs like dots or commas. There are many of them in the manuscript.

Good work.

Author Response

Lines 84-85. Could you add a manufacturer (Company and a country) of the Babbitt alloy and a carbon steel? Where did you get quantitative composition of alloys? Has it come from a manufacturer already?

Company and country has been given in the test of manuscript along with the description of composition.

Lines 92-93. What was the speed of heating (from room temperature to 375°C, etc.) /cooling (from 375°C to 330°C) stages?

The speed of heating/cooling has been added in the text of the manuscript.

Lines 99-100. Please, indicate parts of your Figure 1, as a, b and c. Add scales to the images or indicate sizes, where it is possible.

Parts of Figure 1 have been indicated and scales added wherever possible.

Line 118. What are the concentrations of acids? Were they concentrated? Please, specify it.

Concentrations of acids have been added in the text of manuscript.

Line 121. What is the manufacturer and the model of the electron microscope? Please add this information.

Manufacturer and model of electron microscope has been mentioned in the text of manuscript.

Line 122. Should be. “Interface … was observed…”

Corrected as advised.

Line 123. Should be a comma instead of a dot after the word “Moreover”

Comma has been added after “Moreover” instead of full stop.

Line 155. Horizontal composition of images would be more appropriate.

Images in Figure 3 have been arranged horizontally.

Lines 157-158. The scales on images and numbers, as well as elements on graphs are not readable. Please change the quality of this Figure. Otherwise, it is not clear what is difference between images a) and c), as both of them related to tin-based Babbitt alloy.

The scales in the images have been increased, as suggested.

Line 164. Please, add word “correspondingly” in the end and verify the letters – a, b, c, and d in th text and in the figure caption (line 161). it seems as an error, i.e. should be “Figure 5b, c and d, correspondingly”.

Corrected as advised.

Line 172. Should be “… did not dispersed…”

Corrected, please.

Line 184. What are the loading fractions: 0.25, 0.5 and 1.0 wt.% or 2.5, 5.0 and 1.0 wt.%? Is it a mistake?

Loading fractions have been corrected.

Line 188. What is in the image c)? Please, correct the caption. The same question about loading fraction as for Figure 6, is it 5.0 wt.% or 0.5 wt.%?

Corrected, please.

Figure 6 and 7. The scale bars on SEM images are very small. Please, increase them or modify the images.

The scales in Figures 6 and 7 have been magnified.

Line 200. Please, explain better from where goes that Al and O content is bigger in the Babbitt matrix than in the Cu6Sn5 phase? Is it possible to add a table with elements content?

Explained in the text of manuscript.

Line 202 and 203. Some spaces are missed between the words.

Spaces have been inserted between the words.

Line 207. Figure 8 has very bad scale bar. Please, improve it as well as EDS graphs, where the chemical elements are seen with difficulties. Add explanation for c) in the figure caption. Is b) is for the area 1 and c) for the area 2?

Scale bar has been magnified along with other corrections.

Line 254. Figure 12. The same problem of scale bars and chemical elements labels. It should be definitely changed to better ones. Please, indicate the correspondence between e) and f) and areas of EDS analyses.

Scale bar has been magnified along with other corrections.

Lines 257 and further. What was the need for EDS analysis of tin-based Babbitt alloy here? These results are not discussed in the text. Please, describe them or remove.

Done as suggested.

Plese, add spaces between numbers and words, also after some braquetes of references and punctuation signs like dots or commas. There are many of them in the manuscript.

Corrected as suggested, please.

Round 2

Reviewer 3 Report

Comments are corrected and I think the article can be accepted for publication.